# HIV and cART-Associated Dyslipidemia Among HIV-Infected Children

**DOI:** 10.3390/jcm8040430

**Published:** 2019-03-28

**Authors:** Birkneh Tilahun Tadesse, Byron Alexander Foster, Adugna Chala, Tolossa Eticha Chaka, Temesgen Bizuayehu, Freshwork Ayalew, Getahun H/Meskel, Sintayehu Tadesse, Degu Jerene, Eyasu Makonnen, Eleni Aklillu

**Affiliations:** 1Department of Pediatrics, College of Medicine and Health Sciences, Hawassa University, Hawassa 1560, Ethiopia; 2Division of Clinical Pharmacology, Department of Laboratory Medicine, Karolinska Institute, Karolinska University Hospital Huddinge, 141 86 Stockholm, Sweden; eleni.aklillu@ki.se; 3Departments of Dermatology and Pediatrics, Oregon Health & Science University, Portland, OR 97239, USA; bafoster2@hotmail.com; 4Department of Pharmacology, College of Health Sciences, Addis Ababa University, Addis Ababa 9086, Ethiopia; adugnadema@gmail.com (A.C.); ssintayehu2010@gmail.com (S.T.); eyasumakonnen@yahoo.com (E.M.); 5Adama General Hospital and Medical College, Adama 84, Ethiopia; tecb2006@gmail.com; 6School of Laboratory Medicine, College of Medicine and Health Sciences, Hawassa University, Hawassa 1560, Ethiopia; temesgenbizuayehu2@gmail.com (T.B.); workfresh@gmail.com (F.A.); getchoh326@gmail.com (G.H/M.); 7Management Sciences for Health, Addis Ababa 1250, Ethiopia; degujerene@gmail.com; 8CDT Africa, College of Health Sciences, Addis Ababa University, Addis Ababa 9086, Ethiopia

**Keywords:** dyslipidemias, children, HIV, cART

## Abstract

Background: Persistent dyslipidemia in children is associated with risks of cardiovascular accidents and poor combination antiretroviral therapy (cART) outcome. We report on the first evaluation of prevalence and associations with dyslipidemia due to HIV and cART among HIV-infected Ethiopian children. Methods: 105 cART naïve and 215 treatment experienced HIV-infected children were enrolled from nine HIV centers. Demographic and clinical data, lipid profile, cART type, adherence to and duration on cART were recorded. Total, low density (LDLc) and high density (HDLc) cholesterol values >200 mg/dL, >130 mg/dL, <40 mg/dL, respectively; and/or, triglyceride values >150 mg/dL defined cases of dyslipidemia. Prevalence and predictors of dyslipidemia were compared between the two groups. Results: prevalence of dyslipidemia was significantly higher among cART experienced (70.2%) than treatment naïve (58.1%) children (*p* = 0.03). Prevalence of low HDLc (40.2% versus 23.4%, *p* = 0.006) and hypertriglyceridemia (47.2% versus 35.8%, *p* = 0.02) was higher among cART experienced than naïve children. There was no difference in total hypercholesterolemia and high LDLc levels. Nutrition state was associated with dyslipidemia among cART naïve children (*p* = 0.01). Conclusion: high prevalence of cART-associated dyslipidemia, particularly low HDLc and hypertriglyceridemia was observed among treatment experienced HIV-infected children. The findings underscore the need for regular follow up of children on cART for lipid abnormalities.

## 1. Introduction

According to the UNAIDS 2016 report, approximately 2.1 million children and adolescents below the age of 15 live with HIV worldwide. The report also presents an estimated new infection rate of about 160,000 per year and 120,000 HIV/AIDS-associated deaths annually [1]. Scale up of combination antiretroviral therapy (cART) has successfully reduced HIV/AIDS related morbidity and mortality by well over half [1]. These achievements can be sustained through optimization of treatment including prevention and diagnosis of treatment failure, and minimization of the untoward effects of cART. Even though almost all HIV drugs composing cART have untoward effects [2], the benefits of cART in decreasing AIDS related mortality and morbidity outweigh the risks for cART associated adverse events [1,2].

The second 90 in the 90-90-90 goal is initiating cART for at least 90% of HIV-infected patients by 2020 [3], though there are recent critiques that these goals are neither realistic nor inclusive [4]. There is progress towards achieving the goals by the set deadline for most sub-Saharan African countries [1]. According to the UNAIDS 2017 report, Ethiopia has achieved 67% of HIV diagnosis from total population infected, 88% of initiating treatment for HIV-infected, and 86% of virological suppression among those who started cART [1].

Acute and long-term adverse effects of cART may compromise drug efficacy and consequently decrease survival of HIV-infected children. Metabolic complications including lipodystrophy, lactic acidosis, insulin resistance and dyslipidemia are well established adverse events associated with long term cART use in both children and adults [5], though the burden among HIV-infected children in developing countries has not been well studied [6]. In an Austrian study among adults, dyslipidemia was reported in 46.3% and a significant association was observed with protease inhibitor use [7]. Studies among children have reported dyslipidemia of 38%, and lipodystrophy of 80%; though no significant clinical predictors were observed [8]. A significant increment in the level of insulin resistance among children on long-term cART has been observed among those taking Zidovudine (AZT) and Abacavir (ABC) based regimens [9], with an overall prevalence of 10–15% [9,10]. As cART becomes a long-term intervention to prevent HIV associated morbidity and mortality, understanding the burden of metabolic derangements would allow for development of better prevention and care guidelines for the specific population.

The magnitude of cardiovascular accidents among HIV-infected children because of treatment associated metabolic derangements and/or HIV infection is poorly understood. The real burden of cerebrovascular accidents in children is underestimated due to lack of diagnostic imaging in high HIV prevalence settings, misdiagnosis as HIV encephalopathy and failing to recognize subtle, non-motor manifestations [11]. However, both HIV infection and cART exposure were reported to be risk factors for cerebrovascular accidents [12,13]. A study which included 64 HIV-infected children showed a high prevalence of vasculopathy (38%) and stroke soon after initiation of treatment (25%) [14]. Overall, the evidences suggest that both HIV infection and cART are additional risks to cerebrovascular events in children.

Understanding the magnitude and associations of dyslipidemia and its relationship with treatment status in the HIV-infected pediatric population living in resource limited countries would help to facilitate scale up of cART in the setting. The current study was designed to address two important issues i.e, comparing the lipid profile abnormalities between treatment experienced and naïve HIV-infected children; and, to assess the factors associated with lipid profile abnormalities among cART experienced and naïve HIV-infected children. The study explored the differences in lipid profile abnormalities and identify important clinical and demographic factors associated with dyslipidemia among the two groups–cART naïve and experienced HIV-infected children.

## 2. Methods Section

This study uses data collected in two prospective pediatric HIV cohorts: (1) the EPHIC cohort [15] which was conducted in the Southern Nations Nationalities and Peoples Region (SNNPR) of Ethiopia. EPHIC enrolls cART naive and experienced children. (2) The Efavirenz Pediatric Dose Optimization Study (EPDOS) which enrolls only cART naïve HIV-infected children at HIV treatment centers in two regions; SNNPR and Oromia regional state. Baseline clinical and laboratory data from both cohorts were used for this analysis.

In 2017, there were a total of 6221 HIV-infected children below 14 years of age in the region, with 122 new pediatric infections per year [16]. At the study sites, pediatric HIV treatment services are provided by trained health professionals in hospitals and larger health centers. Study participants were enrolled over six months across six established HIV/AIDS centers with the highest patient volumes in SNNPR–Hawassa University; Yirgalem General Hospital; Arba Minch Hospital; Hossana General Hospital; Adare General Hospital; and Wolayta Sodo Referral hospital; and in Oromia regional state–Adama Referral Hospital, Assela General Hospital, and Shashemene General Hospital.

### 2.1. Study Design and Inclusion Criteria

The current comparative cross sectional study included two groups of HIV-infected children based on treatment status–children who were on cART for at least 6 months (from the EPHIC cohort); and, treatment naïve HIV-infected children (from the EPDOS cohort). EPDOS is a prospective cohort study which enrolls treatment naïve HIV-infected children who are 3–18 years of age. Children who are critically sick, did not consent and had a TB co-infection were excluded from the study. Baseline study participant characteristics including inclusion and exclusion criteria to the EPHIC cohort have been published previously [15]. In brief, EPHIC is a prospective cohort study of children under 18 years with a diagnosis of HIV based on WHO guidelines [17], who were already on first line cART. Children who discontinued follow up before six months on cART, were on second line cART, or had cART failure at enrollment were excluded. For the purpose of the current study, children who were on cART for at least 6 months were selected. From both EPHIC and EPDOS, enrolment data were used to compare the two groups.

### 2.2. Data Collection Procedures

At enrollment, data on demographics, age at HIV diagnosis and cART initiation, nutritional status, WHO clinical stage, hemoglobin, CD4 count/percent, lipid profile parameters–total cholesterol, triglyceride, low density lipoprotein (LDL) cholesterol, and high density lipoprotein (HDL) cholesterol, and plasma viral load were collected for both groups; cART naïve HIV-infected children (EPDOS) and cART experienced children (for at least 6 months) (EPHIC). Clinical, sociodemographic and laboratory data were collected in both groups.

#### 2.2.1. Clinical Data

An assessment for signs and symptoms was completed using a structured symptom checklist. Growth parameters were assessed using WHO growth curves [18], along with assessment of age appropriate developmental milestones [19] and visual analogue scale, VAS adherence [20].

#### 2.2.2. Laboratory Data

Hematologic parameters (complete blood count, CBC, CD4 count (and percentage), total cholesterol, triglycerides, LDL cholesterol, HDL cholesterol, and plasma viral load were determined. Any decision to switch cART or other treatments was solely made by the child’s usual health care provider.

### 2.3. Lipid Profile Assessment and Case Definitions

Non-fasting total cholesterol, low density lipoprotein cholesterol, LDLc, high density lipoprotein cholesterol, HDLc cholesterol and triglycerides were determined while children came for follow up to the HIV clinics. Lipid profile abnormalities were defined based on pediatric references [21]. Accordingly, taking the highest lipid profile level across age groups, total cholesterol values greater than 200 mg/dL; triglyceride level of greater than 150 mg/dL; LDLc level of greater than 130 mg/dL and an HDLc value less than 40 mg/dL were considered as abnormal lipid profile levels.

### 2.4. Statistical Analysis

Descriptive statistics were presented using mean (standard deviation-SD), median (inter quartile rage-IQR), and frequency (percentage). Total cholesterol, triglyceride, LDLc and HDLc values were dichotomized to abnormal or within normal limits (WNL). The association between abnormal lipid profile values and independent variables was assessed using non-parametric tests as the data were found to be non-normally distributed by the Schapiro Wilk test. Groups were compared using Fisher´s exact test for categorical variables and Wilcoxon Rank sum test for continuous variables. To assess for factors which are independently associated with dyslipidemia among cART naïve and experienced HIV-infected children, a multivariate binary logistic regression was done using covariates with a *p*-value < 0.20 in the bivariate analysis. For the multivariate model, a *p*-value < 0.05 was considered as statistically significant. R software 3.5.0(The R Foundation for Statistical Computing, Vienna, Austria) and Graph Pad Prism 7.04 (GraphPad Software, La Jolla, CA, USA) were used for statistical analysis and for graphics.

### 2.5. Data Quality Assurance

Clinical and laboratory data were collected on paper at study sites, the study clinician was charged with checking completeness and clarity every week. Paper forms were then collected by the study team every three months after checking for quality and completeness. The paper forms were entered to REDCap [22], by a central data encoder who completed a quality check of the data for completeness. Viral load, CD4, CBC, lipid profile assessment and other tests were done following standard procedures at each clinical study site.

### 2.6. Ethical Considerations

Ethical approval was obtained from SNNPR Regional Health Bureau Institutional Review Board (IRB) and the National Research and Ethics Review Board of Ethiopia (3-10/46/2018, 21 March 2018). Written informed consent was obtained from the parents/caregivers of children younger than 12 years; while both consent and assent were obtained from children above the age of 12 years. Confidentiality was assured by making all the data collection forms anonymous and securely keeping completed forms. Participation was completely voluntary and participants were free to withdraw from the cohort at any time.

## 3. Results

### 3.1. Sociodemographic Characteristics

A total of 320 HIV-infected children were included in this analysis from both the EPHIC and EPDOS cohorts. The median age was 11 years (interquartile rage (IQR): 7.01–14.0); 165 (49%) of the participants were male. Study participants included two groups: 105 treatment naïve HIV-infected children, and 215 children who were on first line cART. Treatment experienced children were on Zidovudine (AZT), Lamivudine (3TC), Nevirapine (NVP)–90 (39.1%); Stavudine (D4T), 3TC, NVP–54 (23.5%); AZT, 3TC, Efavirenz (EFV)–27 (11.7%); TDF, 3TC, NVP or EFV–15 (6.5%), D4T, 3TC, EFV–12 (5.2%) and Abacavir (ABC), 3TC with one of NVP, 3TC, EFV or a Protease Inhibitors (PI)–4 (1.7%). Treatment experienced children were on cART for a median of 54.0 (IQR: 23.0–86.5) months.

Among treatment naïve children, the WHO clinical status of HIV at enrolment (before cART initiation) was 41 (39.0%) stage 1, 23 (21.9%) stage 2, 31 (29.5%) stage 3, and 9 (8.6%) stage 4. From the cART experienced group, 35 (16.0%), 61 (25.6%), 77 (33.5%) and 11 (5.5%) were WHO clinical stage 1, 2, 3 and 4 at enrolment (with a median duration on cART of 4.5 years), respectively. Despite being on cART, a significantly higher proportion of treatment experienced HIV-infected children had a WHO clinical stage 3 or 4 disease as compared to treatment naïve children (*p* = 0.04).

Treatment naïve children were younger than those who were on cART (*p* < 0.001). The effects of cART on virological suppression and immunologic recovery were evidenced by a significantly lower viral load (*p* < 0.001) and a higher median CD4 count (*p* < 0.001) among treatment experienced children as compared to those who are cART naïve. Similarly, anthropometrics were better among cART experienced children as compared to those who were naïve–HAZ score (*p* = 0.014), WAZ score (*p* < 0.001), and BAZ score (*p* = 0.022). (Table 1). A significantly higher median hepatic enzyme, blood urea nitrogen (BUN) and creatinine values were observed among treatment naïve children as compared to those who are experienced (Table 1).

### 3.2. Lipid Profile Abnormalities among cART Naïve and Experienced Children

The median total cholesterol among cART naïve children was significantly higher than treatment experienced children (median = 120.0, IQR: 97.8–150.0 versus median = 102.5, IQR: 80.0–127.8 respectively) (*p* = 0.001). On the other hand, the median HDL cholesterol was significantly lower among cART experienced children as compared to treatment naïve participants (median = 45.0, IQR: 30.3–48.0 and median=48.5, IQR: 41.0–66.9 respectively) (*p* = 0.002). Similarly, a significantly higher proportion of cART experienced children had HDL levels lower than lower limit of normal (LLN) as compared to those who are treatment naïve (23.4% versus 40.2%, *p* = 0.006). Treatment experienced children also had a higher median triglyceride as compared to cART naïve children, even though it did not reach statistical significance (*p* = 0.16). Comparing the proportion of any lipid abnormality between the two groups, a significantly higher proportion of cART experienced children had at least one lipid abnormality as compared to those who are treatment naïve (70.2% versus 58.1% *p* = 0.033) (Table 2).

### 3.3. Predictors of Lipid Abnormalities among cART Experienced and Naïve HIV-Infected Children

Treatment naïve HIV-infected children with dyslipidemia had poorer anthropometric indices as compared to those who are cART experienced. The median HAZ, WAZ and BAZ were lower among cART naïve children with dyslipidemia as compared to their counterparts (Table 3). On multivariate analysis, only HAZ score was independently associated with dyslipidemia among cART naïve HIV-infected children (Table 4).

Treatment duration appeared shorter among cART experienced children with dyslipidemia as compared to those without dyslipidemia (median = 49, IQR: (21.0–78.0) months versus median = 60, IQR: (30.25–90.75) months), even though it did not reach statistical significance (*p* = 0.253). However, as shown in Figure 1, median total cholesterol and median HDLc levels showed significant difference by cART duration. The median total cholesterol among treatment experienced children who had been on cART for more than 5 years was significantly lower than that in treatment naïve HIV-infected children (*p* = 0.003). On the other hand, the median HDLc among cART experienced children was lower than the median HDLc among treatment naïve HIV-infected children–for children <2 years on cART (*p* = 0.009), those 2–5 years on cART (*p* = 0.005) and more than 5 years on cART (*p* = 0.04). (Figure 1)

## 4. Discussion

In this comparative cross sectional study, both study groups–treatment experienced and cART naïve HIV-infected children–exhibited a high prevalence of lipid profile abnormalities. Interestingly a significantly higher prevalence of lipid profile abnormalities among children and adolescents who were on cART as compared to treatment naïve HIV-infected children was observed. Malnutrition was a significantly associated with dyslipidemia in cART naïve children, but in cART experienced none of the sociodemographic or baseline clinical parameters predicted dyslipidemia. Duration on cART was associated total and HDL cholesterol values as compared to being treatment naïve. To our knowledge this is the first study to explore and compare the prevalence and risk factors of dyslipidemia in cART naïve and cART experience children from sub Saharan Africa.

While chronic medical problems like HIV and obesity could be associated with dyslipidemia, normal sub Saharan African children exhibit a very low prevalence of dyslipidemia [23]. The high prevalence of lipid abnormalities among HIV-infected patients was similarly reported by studies from in different parts of the world [8,24,25,26]. In the current study, the high prevalence of hypertriglyceridemia (35.8% and 47.2% among cART naïve and experienced children, respectively) and low HDLc values (23.4% and 40.2% among cART naïve and experienced children, respectively) were similar to previous reported figures from India among cART naïve HIV-infected children [25], both cART naïve and experienced HIV-infected children from India [26] and treatment experienced HIV-infected children from Uganda [24]. An adult study which was done in the same study setting reported higher lipid profile abnormalities among cART experienced HIV-infected adults as compared to cART naïve counter parts [27]. Our finding of a significantly lower median total cholesterol among cART experienced children might indicate that only specific lipid type abnormalities, for instance abnormalities in triglycerides and HDLc are more common with treatment experience. However, as there is limited evidence in this regard, more studies exploring the specific types of lipid abnormalities would be important.

Extremes of nutritional status and obesity have been reported to be associated with dyslipidemia among children without HIV infection [28,29,30]. In the current study, nutrition state was associated with dyslipidemia among treatment naïve HIV-infected children but was not associated with dyslipidemia among cART experienced children. This could be explained by the beneficial effects of cART leading to the anthropometric improvement observed among HIV-infected children who have been on treatment for at least six months.

Our findings that lipid profile abnormalities are more common among cART experienced children and the finding of lower HDLc levels with increasing duration of cART underscore the need for regular biochemical assessment of HIV-infected children. With the increasing longevity of HIV-infected children and adolescents who are taking more efficacious cART in resource limited settings in Africa, minimizing the long term side effects is very important. Persistent dyslipidemia in children has been linked with cardiovascular disease (CVD) risks [31,32], which could potentially jeopardize the achievements in reducing HIV associated child mortality and morbidity [33]. With regular screening integrated in the HIV care and treatment, adopting recommended life style modifications and lipid lowering treatment in a small proportion would help to decrease the future risks of CVD risks [34].

The untoward effects of persistent dyslipidemia in the pediatric population have been studied among children with renal problems [14]. In this report, it has been shown that children with persistent dyslipidemia could be at risk of cardiovascular and neurologic abnormalities if they are not treated with lipid lowering agents during the early stages of abnormality. The findings of the current study indicate that there is a significantly higher prevalence of dyslipidemia among children on cART as compared to naïve HIV-infected children. Hence, the findings suggest that there is a need for continued follow up and also the need for timely institution of interventions for persistent dyslipidemia.

Our study has several strengths including enrolling both cART naïve and experienced HIV-infected children in a resource limited setting where there is limited follow up for virologic suppression and monitoring for metabolic derangements. However, as the study included children with a wide age range, it was difficult to ensure overnight fasting in young children, which led to the decision of determining lipid profile levels on non-fasting samples in all included children. Moreover, due to logistical reasons, CD8 levels were not determined and we could not assess the effect of CD8 and CD4/CD8 ratio values on lipid profile abnormalities.

## 5. Conclusions

In conclusion, our findings indicate that there is a high prevalence of hypertriglyceridemia and low HDLc dyslipidemia among cART naïve and experienced HIV-infected children and adolescents. Prevalence of any dyslipidemia was significantly higher among cART experienced HIV-infected children as compared to their cART naïve counterparts. The findings underscore the need for monitoring children for untoward effects of long term cART, including dyslipidemia. Children with persistent dyslipidemia will require recommended life style medications, and even in rare instances, lipid lowering drugs to reduce risks of cardiovascular accidents as children or young adults.

## Figures and Tables

**Figure 1 jcm-08-00430-f001:**
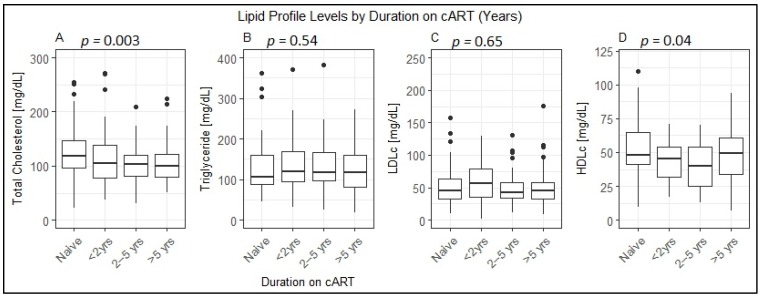
Median (interquartile range) values of lipid profile levels among treatment naïve and antiretroviral therapy experienced HIV-infected children with varying duration of treatment, 2016–2018. Figure 1 depicts the lipid profile levels by duration on cART. (**A**) shows total cholesterol levels by duration on treatment. The median total cholesterol of children who have taken cART for more than 5 years was significantly lower than cART naïve children (*p* = 0.003); children who have been on treatment for less than 2 years and those who have taken treatment for 2–5 years had lower total cholesterol values, but both were not statistically significant (*p* > 0.05); (**B**) shows the triglyceride levels by treatment duration. There was no statistically significance difference in the median triglyceride level by duration of treatment (*p* > 0.05); (**C**) shows LDLc (low density lipoprotein cholesterol) levels by treatment duration. There was no statistically significant difference in the median LDLc values among the groups by treatment experience (*p* > 0.05); (**D**) shows HDLc (high density lipoprotein cholesterol) by treatment duration. Children who have taken cART for more than 2 years have a significantly lower median HDLc level than those who are treatment naïve (*p* = 0.04). yrs–years.

**Table 1 jcm-08-00430-t001:** Sociodemographic characteristics of HIV-infected children who are naïve to antiretroviral therapy and those who have experienced cART for at least six months, 2016–2018.

Variable	cART Naïve (*n* = 105)	cART Experienced (*n* = 215)	*p*-Value
**Male sex**	52.0%	50.5%	0.185 ^†^
**Age, median (IQR)**	9.0(5.0–12.5)	12.0(8.1–14.1)	<0.001 ^‡^
**Age at HIV diagnosis, median(IQR)**		6.10(4.0–10.0)	
**HAZ, median (IQR)**	−1.46(−2.41–(−0.66))	−1.27(−1.99–(−0.21))	0.014 ^‡^
**WAZ, median (IQR)**	−1.41(−2.51–(−0.80))	−0.80(−1.50–0.12)	<0.001 ^‡^
**BAZ, median (IQR)**	−1.16(−2.45–(−0.29))	−0.95(−1.53–(−0.09))	0.022 ^‡^
**Stage 3 or 4 WHO clinical stage**	38.5%	50.0%	0.040 ^†^
**Log10 pVL, median(IQR)**	4.26(3.33–5.01)	2.99(2.36–3.77)	<0.001 ^‡^
**Detectable pVL (>150 copies/mL) ***	95 (100)	53(33.1)	<0.001 ^†^
**CD4 Count, Median (IQR), cells/mL**	330(217–719)	754(518–1050)	<0.001 ^‡^
**Creatinine, Median (IQR), mg/dL**	0.60(0.40–0.70)	0.60(46–0.70)	0.876 ^‡^
**BUN, Median (IQR), mg/dL**	17.0(12.10–24.50)	9.0(4.30–15.0)	<0.001 ^‡^
**ALT, Median (IQR), units/L**	26.0(19.0–36.5)	21.0(16.0–32.0)	0.003 ^‡^
**AST, Median (IQR), units/L**	39.0(32.2–53.0)	32.0(24.5–41.0)	<0.001 ^‡^
**Hemoglobin, Median (IQR), mg/dL**	12.3(11.4–13.2)	13.4(12.5–14.5)	<0.001 ^‡^

^‡^ Mann-Whitney test used to compare the medians; ^†^ Fisher’s Exact test was used; cART–combination antiretroviral therapy; BUN–Blood Urea Nitrogen; ALT–Alanine Aminotransferase; AST–Aspartate Aminotransferase; HAZ–Height for Age Z-core; WAZ–Weight for Age Z-core; BAZ–Body Mass Index for Age Z-core; WHO–World Health Organization; IQR–Interquartile range; * Ten children from the cART naïve group had insufficient sample and pVL could not be done.

**Table 2 jcm-08-00430-t002:** Lipid profile values among children who are antiretroviral therapy naïve and those who have been on antiretroviral therapy for at least six months, 2016–2018.

Lipid Parameter	cART Naïve (*n* = 105)	cART Experienced (*n* = 215)	Odds Ratio (95% Confidence Interval)	*p*-Value
**Total Cholesterol, median (IQR) mg/dL**	120 (97.8–150.0)	102.5 (80.0–127.8)		0.001 ^‡^
**Total Cholesterol >200 mg/dL**	6.2%	3.3%	0.56(0.18–1.70)	0.233 ^†^
**Triglyceride, median (IQR) mg/dL**	106.0 (88.5–162.0)	127.5 (96.3–169.8)		0.165 ^‡^
**Triglyceride level >150 mg/dL**	35.8%	47.2%	1.45(1.06–1.98)	0.015 ^†^
**HDLc, median (IQR) mg/dL**	48.5 (41.0–66.9)	45.0 (30.3–48.0)		0.002 ^‡^
**HDLc level <40 mg/dL**	23.4%	40.2%	2.42(1.40–4.17)	0.006 ^†^
**LDLc, median (IQR) mg/dL**	45.0 (33.0–64.0)	46.1 (32.0–68.9)		0.872 ^‡^
**LDLc level >130 mg/dL**	2.9%	2.0%	1.41(0.29–6.84)	1.000 ^†^
**Any dyslipidemia**	58.1%	70.2%	3.27(1.32–2.13)	0.033 ^†^

^‡^ Mann-Whitney test used to compare the medians; ^†^ Fisher’s Exact test was used; cART–combination antiretroviral therapy; HDLc-high density lipoprotein cholesterol; LDLc-low density lipoprotein cholesterol.

**Table 3 jcm-08-00430-t003:** Predictors of dyslipidemia among antiretroviral therapy naïve and experienced HIV-infected children, 2016-2018.

Variable	cART Naïve		*p* Value	cART Experienced		*p* Value
	Dyslipidemia	No Dyslipidemia		Dyslipidemia	No Dyslipidemia	
**Age in years, median (IQR)**	9.0(5.0–12.0)	9.0(6.0–12.75)	0.488 ^‡^	12(7.77–14.07)	13(9–14.8)	0.181 ^‡^
**Sex, N (%) Male**	33(54.1)	29(65.9)	0.236 ^†^	75(51.7)	28(47.5)	0.644 ^†^
**Log10 pVL, median (IQR)**	4.22(3.38–5.0)	4.28(3.17–5.0)	0.641 ^‡^	2.83(2.2–4.0)	3.03(2.62–3.63)	0.604 ^‡^
**cART duration, Median (IQR), months**				49(21.0–78.0)	60(30.25–90.75)	0.253 ^‡^
**Median (IQR) Hgb in gm/dL**	12.50(11.03–13.45)	12.05(11.58–13.03)	0.814 ^‡^	13.3(12.3–14.3)	13.7(12.7–15.0)	0.063 ^‡^
**CD4, Median (IQR), cells/dL**	446(263–794)	319(217–447)	0.241 ^‡^	775(543–1060)	628(465–1023)	0.234 ^‡^
**WHO stage, N (%) Stage 3 or 4**	27(45.0)	13(29.5)	0.153 ^†^	63(52.1)	25(45.5)	0.516 ^†^
**Creatinine, Median (IQR), mg/dL**	0.53(0.41–0.70)	0.60(0.43–0.75)	0.393 ^‡^	0.6(0.45–0.70)	0.6(0.43–0.70)	0.744 ^‡^
**BUN, Median (IQR), mg/dL**	16.0(12.0–22.0)	18.5(14.25–27.75)	0.283 ^‡^	9(4.0–15.0)	8.37(5.0–15.0)	0.866 ^‡^
**ALT, Median (IQR), units/L**	25.0(20–38.0)	27.0(18.5–36.0)	0.830 ^‡^	21.0(16.0–32.0)	22.0(16–30.3)	0.926 ^‡^
**AST, Median (IQR), units/L**	39.0(34.0–52.25)	39.5(31.25–52.5)	0.841 ^‡^	32.0(25.0–41.0)	32.0(24.0–40.8)	0.610 ^‡^
**NRTI class**						0.332 ^†^
**N (%) Zidovudine**				80(53.0)	27(42.2)	
**N (%) Stavudine**				40(26.5)	24(37.5)	
**N (%) Tenofovir**				9(6.0)	2(3.1)	
**NNRTI class**						0.644 ^†^
**N (%) Efavirenz**				27(17.9)	9(14.1)	
**N (%) Nevirapine**				93(61.6)	42(65.6)	
**HAZ, Median (IQR)**	−1.64(−2.62–(−0.90))	−1.21(−1.83–(−0.45)	0.041 ^‡^	−1.26(−2.0–(−0.21)	−1.29(−1.92–(−0.22))	0.921 ^‡^
**WAZ, Median (IQR)**	−1.86(−2.87–(−1.0))	−1.05 (−1.57–(−0.54))	0.014 ^‡^	−0.82(−1.58–0.01)	−0.44(−1.33–0.42)	0.332 ^‡^
**BAZ, Median (IQR)**	−1.33(−2.67–(−1.55))	−1.04(−2.0–0.04)	0.104 ^‡^	−0.89(−1.44–(−0.09)	−1.05(−1.86–(−0.14)	0.429 ^‡^

^‡^ Mann-Whitney test used to compare the medians; ^†^ Fisher’s Exact test was used; cART–combination antiretroviral therapy; pVL–plasma viral load; IQR–interquartile range; Hgb–Hemoglobin; BUN–Blood Urea Nitrogen; ALT–Alanine Aminotransferase; AST–Aspartate Aminotransferase; NRTI–Nucleoside Reverse Transcriptase Inhibitor; NNRTI–Non Nucleoside Reverse Transcriptase Inhibitor; HAZ–Height-for-Age Z-core; WAZ–Weight for-Age Z-core; BAZ–Body Mass Index-for-Age Z-core.

**Table 4 jcm-08-00430-t004:** Multivariate logistic regression analysis of variables associated with dyslipidemia among cART naïve and experienced HIV-infected children.

Variables	Dyslipidemia Among cART Naïve aOR (95% CI)	*p*-Value	Dyslipidemia Among cART Experienced aOR (95% CI)	*p*-Value
Age in years			0.96(0.88–1.04)	0.279
Hemoglobin in mg/dL			0.97(0.88–1.06)	0.458
Stage 3 or 4 WHO Clinical Stage	0.63(0.17–2.27)	0.478		
Weight for age Z-score	0.64(0.38–1.06)	0.082		
Height for age Z-score	0.74(0.55–0.99)	0.049		

aOR–adjusted odds ratio; 95% CI–95% confidence interval; Body Mass Index (BMI) Z-score (with *p*-value = 0.104 in the bivariate analysis) was not included in the multivariate model to avoid collinearity.

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
