# Peer review of "HIV and cART-Associated Dyslipidemia Among HIV-Infected Children"

_jcm, 2019, doi:10.3390/jcm8040430_

Round 1
Reviewer 1 Report
Thank you for the opportunity to review this cross-sectional study of a multi-site cohort of paediatric patients (under 18 years of age) living with HIV in Ethiopia. The study compared cART-naïve (n=105) and first-line cART-treated (for at least 6 months) (n=215) to determine prevalence and predictors of dyslipidemia. The authors reported that the prevalence of dyslipidemia was significantly higher among the cART-treated than the cART-naïve.
Comments
I think the authors should re-consider their use of the term “predictors” as this was not a longitudinal study. As the data is collected cross-sectionally, I would encourage them to use the term “associations” as there is no way of determining causalty (which is what the term “predictors” denotes).
Abstract
Line 37-38: references ‘cART-associated dyslipidemia’ however the statement made is “observed among HIV infected children” so it is unclear whether the treatment condition was associated or not in that final statement
Introduction
Line 62: in the body of texts this indicates an Australian study, however, the reference reports an Austrian study (typo)
Comment: I wonder if you would consider adding any prevalence data for associated harms? For example, in the abstract, specific reference is made to the risk of cerebrovascular accident (CVA). Are there data on the frequency of CVA in paediatric patients living with HIV?
Methods
Line 122: statistical analysis – did the authors attempt to do a multivariate analysis (MVA) on the data (i.e. to determine if some of the “predictors” [associations] were still statistically significant given the confounders / controlling for the other “predictors” [associations]). In this section they have not outlined any MVA. I think this would strengthen the results of the paper (in addition to what has already been done here, not replacing it).
Line 134: I think a reference for REDCap should be added
Results
Line 150: Although there is an assumption that your readership are from the field, the generic names should be provided for antiretroviral drugs (as opposed to the use of acronyms in the first instance)
Line 154-159: Please make it clearer, were all enrolments to the cohort at diagnosis? If not, then are you indicating that the WHO stages are current? Or at enrolment? If at enrolment, how long treated? Or not yet treated? (i.e. maybe an over-reliance on stating that the cohort has been “published elsewhere”. Would be helpful to still include some of those data here)
Table 1: consider adding a new row of “detectable / undetectable viral load” and also, mean age at diagnosis (or alternatively, mean time living with HIV)
Line 189: as all results were collected at baseline (a cross-sectional analysis), please refrain from the term “predictors of” and change to “associations with”
Line 197: “Median total cholesterol among treatment experienced children who had been on cART for more than 5 years was significantly lower than that in the treatment naïve HIV infected population”
· This requires discussion – what’s happening here? Was it because the children who were cART-naïve were viraemic? Even when you break out the cART-treated only, those with less treatment time (i.e. shorter treatment duration – line 193) were more likely to have dyslipidemia. Again, were these children more recently viraemic? Difficult to decipher because we weren’t given clear data about whether all children were enrolled with viraemic control (although one earlier sentence hinted that an exclusion criterion was “failure at baseline”)
Table 3: this is great, but would you consider also providing an adjusted model? (that is, a MVA?)
Discussion
Line 230: instead of “risk factors” please use “associated factors” or “factors associated with”
Line 249: “Our findings that lipid profile abnormalities are more common among cART experienced children and the finding of more lipid abnormality with increasing duration of cART underscore the need regular biochemical assessment of HIV infected children.” Is this not the opposite if what was stated in Line 197?
Line 271: Have I missed this previously, that non-fasting samples were used? Please ensure this is included in the methods.
I think that this is an important paper that should be modified as per the above to strengthen it for publication. I commend the authors on their work thus far. I think that with some additional attention this work merits publication.
Author Response
Response to Reviewer 1 Comments
Thank you very much for the constructive comments. Below are our responses to teh reviewer's comments.
Point 1: I think the authors should re-consider their use of the term “predictors” as this was not a longitudinal study. As the data is collected cross-sectionally, I would encourage them to use the term “associations” as there is no way of determining causalty (which is what the term “predictors” denotes).
Response 1: Thank you for the comment. We have changed the “predictors” to “associations” throughout the text.
Abstract:
Point 2: Line 37-38: references ‘cART-associated dyslipidemia’ however the statement made is “observed among HIV infected children” so it is unclear whether the treatment condition was associated or not in that final statement
Response 2: We agree with the comment. We have modified the sentence to reflect treatment status. “…was observed among treatment experienced HIV infected children.”
Introduction
Point 3: Line 62: in the body of texts this indicates an Australian study, however, the reference reports an Austrian study (typo)
Response 3: Thank you for pointing this out. Correction has been made.
Point 4: I wonder if you would consider adding any prevalence data for associated harms? For example, in the abstract, specific reference is made to the risk of cerebrovascular accident (CVA). Are there data on the frequency of CVA in paediatric patients living with HIV?
Response 4: Thank you for the important comment. In the introduction part we have included a paragraph to address the prevalence of cerebrovascular accidents among HIV infected children.
Methods:
Point 5: Line 122: statistical analysis – did the authors attempt to do a multivariate analysis (MVA) on the data (i.e. to determine if some of the “predictors” [associations] were still statistically significant given the confounders / controlling for the other “predictors” [associations]). In this section they have not outlined any MVA. I think this would strengthen the results of the paper (in addition to what has already been done here, not replacing it).
Response 5: Thank you. We agree with the comment and we have included a statement in the statistical analysis part regarding multivariate analysis.
Point 6: Line 134: I think a reference for REDCap should be added
Response 6: Thank you. Citation for REDCap was included.
Results:
Point 7: Line 150: Although there is an assumption that your readership are from the field, the generic names should be provided for antiretroviral drugs (as opposed to the use of acronyms in the first instance)
Response 7: Generic names have been included in the first paragraph of the results section.
Point 8: Line 154-159: Please make it clearer, were all enrolments to the cohort at diagnosis? If not, then are you indicating that the WHO stages are current? Or at enrolment? If at enrolment, how long treated? Or not yet treated? (i.e. maybe an over-reliance on stating that the cohort has been “published elsewhere”. Would be helpful to still include some of those data here)
Response 8: Thank you for the suggestion. We have clarified the wording in the revised manuscript (Line 165-168). cART naïve children were not on treatment and enrolment was around the time of HIV diagnosis. Treatment experienced children were on cART for a median of 4.5 years.
Point 9: Table 1: consider adding a new row of “detectable / undetectable viral load” and also, mean age at diagnosis (or alternatively, mean time living with HIV)
Response 9: Thank you for the suggestion. We have included the two variables in Table 1. As the qPCR machine at our lab has a lower limit of detection of 150 copies/mL, we have taken that cut-off for detectable versus not detectable viral load.
Point 10: Line 189: as all results were collected at baseline (a cross-sectional analysis), please refrain from the term “predictors of” and change to “associations with”
Response 10: Corrections were made.
Point 11: Line 197: “Median total cholesterol among treatment experienced children who had been on cART for more than 5 years was significantly lower than that in the treatment naïve HIV infected population”. This requires discussion – what’s happening here? Was it because the children who were cART-naïve were viraemic? Even when you break out the cART-treated only, those with less treatment time (i.e. shorter treatment duration – line 193) were more likely to have dyslipidemia. Again, were these children more recently viraemic? Difficult to decipher because we weren’t given clear data about whether all children were enrolled with viraemic control (although one earlier sentence hinted that an exclusion criterion was “failure at baseline”)
Response 11: Thank you for pointing this out. Median cholesterol was found to be lower among cART experienced than those who are naïve; but not proportion of patients with total hypercholesterolemia (which is total cholesterol above the ULN). Proportion of patients with total hypercholesterolemia was not significantly different between the groups. While lipid profile abnormalities were significantly more common (i.e proportion of any dyslipidemia, hypertriglyceridemia, and/or low HDLc) among cART experienced children, the median total cholesterol was significantly lower among cART experienced than naïve. We have included a statement in the discussion to address this finding (Line 260-264).
Point 12: Table 3: this is great, but would you consider also providing an adjusted model? (that is, a MVA?)
Response 12: Thank you for the suggestion. We have now added Table 4 which includes a multivariate analysis of factors associated with dyslipidemia in both groups. Using the p-values in the bivariate analysis in Table 3 (taking all variables with p-value < 0.2), we have done a multivariate model as shown in Table 4.
Discussion:
Point 13: Line 230: instead of “risk factors” please use “associated factors” or “factors associated with”
Response 13: Thank you. Done.
Point 14: Line 249: “Our findings that lipid profile abnormalities are more common among cART experienced children and the finding of more lipid abnormality with increasing duration of cART underscore the need regular biochemical assessment of HIV infected children.” Is this not the opposite if what was stated in Line 197?
Response 14: Line 197 addresses the median total cholesterol and not proportion of hypercholesterolemia. However, as is shown in Figure 1, the HDLc values were also decreasing with duration on cART and the prevalence of abnormal HDLc level was significantly higher among cART experienced. We have specified this part of the discussion to reflect the changes in HDLc values.
Point 15: Line 271: Have I missed this previously, that non-fasting samples were used? Please ensure this is included in the methods.
Response 15: Thank you. We have included this in the methods section.
Point 16: I think that this is an important paper that should be modified as per the above to strengthen it for publication. I commend the authors on their work thus far. I think that with some additional attention this work merits publication.
Response 16: Thank you.
Reviewer 2 Report
I do only have a comment. Are data from CD8 T cells collected? CD4/CD8 ratio should be analyzed, study, and discussed due to the relevance of the comment that “children with persistent dyslipidemia could be at risk of cardiovascular and neurologic abnormalities”
Author Response
Response to Reviewer 2 Comments
Thank you very much for the constructive comments. Below are our responses to teh reviewer's comments.
Point 1: I do only have a comment. Are data from CD8 T cells collected? CD4/CD8 ratio should be analyzed, study, and discussed due to the relevance of the comment that “children with persistent dyslipidemia could be at risk of cardiovascular and neurologic abnormalities”
Response 1: Thank you for the comment. We don’t have CD8 data and hence the ratio. We agree with importance of the comment and we have included it in the limitations part (Line 296 – 298).
Reviewer 3 Report
This clinical study provides a substantial evidence of dyslipidemia associated with HIV infection and antiretroviral treatment in the HIV infected children, it indicates that there is a high prevalence of hypertriglyceridemia and low HDLc dyslipidemia among cART naïve and experienced HIV infected children and adolescents. It has many strengths including comparing cART naïve and HIV infected children, with solid data analysis methodology. Only concern is the clinical enrollment did not exclude the participants who were not fast. Overall this study shed light on the necessities of monitoring and providing intervention to the children who encounter the hypertriglyceridemia.
Author Response
Response to Reviewer 3 Comments
Thank you very much for the constructive comments. Below are our responses to the reviewer's comments.
Point 1: This clinical study provides a substantial evidence of dyslipidemia associated with HIV infection and antiretroviral treatment in the HIV infected children, it indicates that there is a high prevalence of hypertriglyceridemia and low HDLc dyslipidemia among cART naïve and experienced HIV infected children and adolescents. It has many strengths including comparing cART naïve and HIV infected children, with solid data analysis methodology. Only concern is the clinical enrollment did not exclude the participants who were not fast. Overall this study shed light on the necessities of monitoring and providing intervention to the children who encounter the hypertriglyceridemia.
Response 1: Thank you for the comment. All the children in the study gave non-fasting samples. As fasting was not possible for most children, we uniformly collected non-fasting samples. However, there is also recent consensus that fasting samples might not be required for lipid profile assessment in children. We have also included it in the limitations part anyway (Steiner et al, Pediatrics, 2011; 128(3): 463-470).
Round 2
Reviewer 2 Report
I do not have any more questions.